# Rhizosphere Microbiome and Phenolic Acid Exudation of the Healthy and Diseased American Ginseng Were Modulated by the Cropping History

**DOI:** 10.3390/plants12162993

**Published:** 2023-08-19

**Authors:** Jiahui Zhang, Yanli Wei, Hongmei Li, Jindong Hu, Zhongjuan Zhao, Yuanzheng Wu, Han Yang, Jishun Li, Yi Zhou

**Affiliations:** 1Ecology Institute of Qilu University of Technology (Shandong Academy of Sciences), Jinan 250013, China; 10431201056@stu.qlu.edu.cn (J.Z.); weiyl@sdas.org (Y.W.); lihm0204@sdas.org (H.L.); hujd@sdas.org (J.H.); zhaozjfrances@163.com (Z.Z.); wuyzh@sdas.org (Y.W.); 10431211096@stu.qlu.edu.cn (H.Y.); 2School of Agriculture, Food and Wine, The University of Adelaide, Glen Osmond, SA 5064, Australia

**Keywords:** American ginseng, phenolic acid, p-coumaric acid, microbial community, pathogen abundance

## Abstract

The infection of soil-borne diseases has the potential to modify root exudation and the rhizosphere microbiome. However, the extent to which these modifications occur in various monocropping histories remains inadequately explored. This study sampled healthy and diseased American ginseng (*Panax quinquefolius* L.) plants under 1–4 years of monocropping and analyzed the phenolic acids composition by HPLC, microbiome structure by high-throughput sequencing technique, and the abundance of pathogens by quantitative PCR. First, the fungal pathogens of *Fusarium solani* and *Ilyonectria destructans* in the rhizosphere soil were more abundant in the diseased plants than the healthy plants. The healthy American ginseng plants exudated more phenolic acid, especially p-coumaric acid, compared to the diseased plants after 1–2 years of monocropping, while this difference gradually diminished with the increase in monocropping years. The pathogen abundance was influenced by the exudation of phenolic acids, e.g., total phenolic acids (r = −0.455), p-coumaric acid (r = −0.465), and salicylic acid (r = −0.417), and the further in vitro test confirmed that increased concentration of p-coumaric acid inhibited the mycelial growth of the isolated pathogens for root rot. The healthy plants had a higher diversity of rhizosphere bacterial and fungal microbiome than the diseased plants only after a long period of monocropping. Our study has revealed that the cropping history of American ginseng has altered the effect of pathogens infection on rhizosphere microbiota and root exudation.

## 1. Introduction

American ginseng (*Panax quinquefolius* L.) is one of the most important medical plants in the world and has been widely cultivated under monocropping to increase productivity. However, this practice has also shown negative impacts on American ginseng’s medicinal quality [1]. Monocropping faces various challenges, from the reduction of microbial diversity in the soil [2], degradation of soil organic matter [3], plant autotoxicity [4], and increased susceptibility to soil-borne diseases [5].

A rhizosphere microbial community, including soil-borne diseases, plays an important role in determining the health of *Panax* plants [6,7,8]. The pathogens for American ginseng root rot include *Fusarium solani*, *F. oxysporum*, and *Cylindrocarpon destructans* [1,9], which can damage the seedling’s roots, and mainly spread through soil moisture, underground insects, and nematodes [10]. The previous study on healthy and diseased *P. notoginseng* showed that the differed composition of bacterial and fungal communities in the rhizosphere soil [7] and the abundance of *Pseudomonas* spp. and *Ilyonectria* spp. in the diseased plants were significantly higher than the healthy plants, confirming the interactions between pathogen diseases and the microbiome [11]. On the other hand, compared with the healthy citrus, the relative abundances of the potentially plant-beneficial bacteria *Bradyrhizobium* spp. and *Burkholderia* spp. were lower in the root of the diseased plant [12]. *Burkholderia* spp. could degrade the fungal cell wall through their hydrolytic and catalytic enzymes [13]. In addition, several ISR (Induced Systemic Resistance)-associated genes in the plant, such as SAM, PR1, and PR2, were induced by *Bradyrhizobium* spp. and *Burkholderia* spp. [14].

In the monocropping system, where the same crop is cultivated repeatedly, there is a risk of decreased diversity of microorganisms in the rhizosphere [15]. This reduction may be caused by nutrient depletion in the soil and the accumulation of pathogenic microorganisms [16]. Studies have demonstrated that monocropping can lead to a decline in the abundance and diversity of the plant growth-promoting microbes in the rhizosphere, e.g., *Burkholderia* spp., *Pseudomonas* spp., and *Trichoderma* spp. [17], which play essential roles in improving plant nutrient uptake, disease suppression, and root development [18]. However, there are few studies evaluating the disease effect on the American ginseng rhizosphere microbial communities across various monocropping histories.

Phenolic acids exudated from plant roots can change the structure of soil microorganisms by stimulating soil-borne pathogens and inhibiting beneficial microbes [6,19,20]. For example, wilt-diseased tomato (*Solanum lycopersicum*) had higher concentrations of p-hydroxybenzoic acid, vanillic acid, and ferulic acid in the rhizosphere soil than the healthy plants, and the increased concentrations of phenolic acids significantly stimulated *F. solani* growth [21]. The exudation of cinnamic acid increased the incidence rates of *Fusarium* wilt disease in faba bean (*Vicia faba* L.) [22]. In the soil planted with *P. notoginseng*, the total amount of phenolic acids increased with the continuous planting years [11]. For American ginseng, the growth of the pathogen species *Rhizoctonia solani* was promoted by the low-level exudation of phenolic acids, e.g., p-coumaric acid, syringic acid, and vanillic acid, but suppressed by the higher concentrations of phenolic acid [23]. Phenolic acid in the rhizosphere soil of American ginseng was changed with the increase in monocropping years and drove the dynamics of the pathogenic microorganisms [24]. Therefore, it is crucial to identify the interactions among diverse phenolic acids, microbiome, and pathogens’ occurrence in the rhizosphere microhabitat of American ginseng.

The objectives of this study were to investigate the variations of phenolic acids and microorganisms in the rhizosphere soil of the healthy and diseased American ginseng plants cultivated under continuous monocropping practices and explore the relationships between the phenolic acids, rhizosphere microbiome structure, and pathogen abundance in the rhizosphere.

## 2. Result

### 2.1. The Abundance of American Ginseng Pathogens in the Rhizosphere Soil

Four common pathogens (*F. solani*, *F. oxysporum*, *A. alternata,* and *I. destructans*) of American ginseng were selected for the qPCR analysis. The difference in pathogens abundance in the healthy and diseased American ginseng was affected by monocropping years. In general, the presence of pathogens in the rhizosphere soil of the diseased American ginseng was either greater or comparable to that of the healthy plants (Figure 1).

The abundance of *F. solani* in the diseased American ginseng was significantly higher than that in the healthy plants with under 2 and 4 years of monocropping (Figure 1A). While under 1 and 3 years of monocropping, the disease infection had no influence on the abundance of *F. solani* in the rhizosphere soil. For the abundance of *F. oxysporum* and *A. alternata* in the rhizosphere soil, there was no difference between diseased and healthy American ginseng (Figure 1B,C). Compared with healthy plants, the *I. destructans* abundance in the rhizosphere soil of diseased American ginseng increased by 30.99% and 36.04% under 1 and 4 years of monocropping history, respectively (Figure 1D), while the difference was not significant under 2 or 3 years of monocropping history.

### 2.2. Phenolic Acid Content in the Rhizosphere Soil of American Ginseng

In total, 12 compounds of phenolic acids were detected by the HPLC system according to their retention time as protocatechuic acid, epicatechin, p-hydroxybenzoic acid, vanillic acid, erucic acid, syringic acid, p-coumaric acid, vanillin, ferulic acid, benzoic acid, salicylic acid, and cinnamic acid (Figure 2A). Compared with the healthy plants, the content of total phenolic acids in the rhizosphere soil of the diseased American ginseng significantly reduced by 15.65% and 27.30% under 1 and 2 monocropping years, respectively. The disease effect was not significant when the monocropping history became longer, 3 and 4 years (Figure 2A). The content of phenolic acid in the rhizosphere soil of American ginseng decreased with the increase in monocropping years.

The p-coumaric acid had the highest relative content among the 12 phenolic acids in the rhizosphere soil of American ginseng, about 40.57% (Figure 2B). The concentration of p-coumaric acid exhibited a gradual decline with increasing years of monocropping, from 42.42% in 1 year of monocropping to 31.84% in 4 years of monocropping. The rhizosphere soil of diseased American ginseng showed a decreased level of p-coumaric acid in comparison to healthy plants (38.88% vs. 41.96%). Likewise, when compared to healthy plants, the benzoic acid content in the rhizosphere soil of American ginseng that had been diseased for 1 year exhibited a significant reduction of 16.30%. However, there were no significant differences in benzoic acid levels between healthy and diseased soil in the 2–4 years of monocropping. The VIP diagram of PLS-DA (Figure 2C) and the random forest analysis (Figure 2D) revealed that p-coumaric acid and syringic acid were the major metabolites driving the variation in phenolic acid composition in the rhizosphere soil of healthy and diseased American ginseng with varied monocropping years (*p* < 0.05).

### 2.3. Microbial Communities in the Rhizosphere Soil of the Healthy and Diseased American Ginseng

The Shannon index was used to indicate the alpha diversity of rhizosphere soil fungal and bacterial microbiome in each sample (Figure 3A,B). For both fungal and bacterial communities, the Shannon index in the rhizosphere soil of healthy American ginseng was higher in comparison to diseased plants only under 4 years of monocropping (Figure 3).

The fungal (Appendix A) and bacterial (Appendix A) microbiome in the rhizosphere soil of healthy and diseased American ginseng during monocropping years were compared using PCoA analysis based on the Bray–Curtis distance. The results showed that the first principal coordinates explained 40.60% and 39.70% of the total variation for the fungal and bacterial microbiomes, respectively. The fungal and bacterial microbiomes in diseased soil after 4 years of monocropping were significantly different from those in other years (Appendix A). The difference in the rhizosphere fungal and bacterial microbiome between healthy and diseased American ginseng was greater after 4 years of monocropping (*p* < 0.05) compared to 1, 2, and 3 years of monocropping (Figure 4A,B).

The dominant fungal phylum of healthy and diseased rhizosphere soil belonged to Ascomycota (17.49–47.48%) (Appendix A). Compared with the healthy American ginseng, the relative abundances of Ascomycota in the rhizosphere soil of diseased plants reduced after 1 and 4 years of monocropping, whereas they increased after 2 and 3 years.

A further annotation at the fungal genus level illustrated that the dominant genera were *Fusarium* spp. (2.10% to 25.88%), *Trichoderma* spp. (0.60% to 31.33%), and *Ilyonectria* spp. (0.03% to 32.65%). The relative abundance of *Trichoderma* spp. in the rhizosphere soil of the diseased plants was lower than that in the healthy American ginseng in 1–4 monocropping years. Compared with the diseased American ginseng, the relative abundance of *Fusarium* spp. of the healthy plant in 1, 2, and 3 years decreased by 3.74%, 18.79%, and 11.73%. In addition, after 4 years of monocropping, *Ilyonectria* spp. in the rhizosphere soil of the diseased plant was highly enriched compared with the healthy plant (0.42% vs. 32.65%).

The Proteobacteria (34.27% to 48.83%) and Actinobacteria (19.07% to 33.74%) were the dominant bacterial phyla in all the samples. The relative abundance of Actinobacteria in the healthy plants was higher than in diseased plants in 1, 2, and 3 years of monocropping (Appendix A). The relative abundance of Actinobacteria gradually decreased with the increase in monocropping years. For bacterial genera, *Candidatus Koribacter* spp., *Bacillus* spp., and *Burkholderia* spp. were dominant in the rhizosphere soil. Compared with the healthy American ginseng, the relative abundance of *Candidatus Koribacter* spp. in rhizosphere soil of the diseased plants increased by 0.90% and 2.24% in 1 and 3 years of monocropping, respectively, but decreased by 0.34% and 1.69% in 2 and 4 years of monocropping, respectively. There was no significant difference between the *Bacillus* spp. and *Burkholderia* spp. between healthy and diseased American ginseng (Figure 4D).

### 2.4. Relationship between Phenolic Acids and Pathogens Abundance

The Pearson correlation was used to analyze the relationship between pathogen abundance and phenolic acids. The results illustrated that the abundance of *F. solani* had a significantly negative correlation with the content of sinapic acid (*p* = 0.026 < 0.05), p-coumaric acid (*p* = 0.022 < 0.05), and total phenolic acids (*p* = 0.025 < 0.05) (Figure 5). Additionally, the *I. destructans* abundance was negatively correlated with salicylic acid (*p* = 0.042 < 0.05) (Figure 5). However, the abundances of *A. alternata* and *F. oxysporum* were not associated with the content of phenolic acids.

The mycelial growth of *F. solani* was significantly inhibited when the total phenolic acid was at a concentration ≥ 80 mg·L^−1^ with response indexes of 3.45% to 8.28% (Figure 6A). *F. solani* was significantly inhibited by p-coumaric acid at ≥40 mg·L^−1^ with response indexes of 3.45% to 7.59% (Figure 6B).

## 3. Discussion

### 3.1. Potential Pathogens for Root Rot of American Ginseng on the Different Monocropping Years

Our study demonstrated that *F. solani* and *I. destructans* were the potential pathogens for American ginseng root rot, indicated by their higher abundances in the rhizosphere soil of diseased plants compared with healthy plants after 4-year monoculture. Previous findings showed that the pathogens that caused root rot in American ginseng mainly included *Fusarium* spp., *Cylindrospora* spp., and *Ilyonectria* spp. [25,26]. *F. solani* was a common plant pathogen that caused root rot and crown rot in potato [27], ginseng [28], and peanut [29]. Our results were consistent with the research in *P. notoginseng* rhizosphere soil, which found that the relative abundance of *Ilyonectria* spp. in 1- and 3-year-old plants was significantly higher than that in the 2-year-old plants, while the relative abundance of *Fusarium* spp. was highest in the rhizosphere soil of the 2-year-old plants [30]. Another study revealed that the relative abundance of *Ilyonectria* spp. in the rhizosphere soil of root rot of *P. notoginseng* was up to 75.90% to 80.10%, but it was not successfully isolated, possibly because of the slow growth characteristics [31].

### 3.2. The Changes in Phenolic Acid in the Healthy and Diseased American Ginseng during Various Consecutive Years of Monocropping

The phenolic acids in the soil were from the decomposition of plant residues and root exudation, which affected the growth and germination of American ginseng seedlings [32]. In our study, the difference in the total phenolic acid between the diseased and healthy American ginseng decreased gradually with the increase in monocropping years. The phenolic acid content in the healthy plants was larger than diseased plants after short-term monocropping, probably because the phenolic acid in the diseased American ginseng was degraded into resistant substances such as lignin [33]. Lignin can strengthen the cell walls and hinder pathogen invasion [34]. After long-term monocropping, there was no difference in phenolic acid content in the rhizosphere soil between healthy and diseased American ginseng, possibly because the root of the diseased plants decayed after a long-term monocropping and released a large quantity of phenolic acid into the soil to compensate the reduction consumed for resisting the pathogen.

The phenolic acids profiles in the rhizosphere soil of both healthy and diseased American ginseng were dominated by p-coumaric acid, which was consistent with the result in *P. notoginseng* [22]. P-coumaric acid is an important phenolic acid in the phenylalanine metabolism pathway of plants as the precursor of lignin synthesis and plays a significant role in pathogen resistance [35]. Islam et al. [36] found that p-coumaric acid promoted the growth of the plant cell wall and induced resistance in Chinese cabbage (*Brassica rapa* var. *pekinensis*) against the pathogens of *Xanthomonas campestris* pv. *Campestris*. Since p-coumaric acid in the rhizosphere soil had the highest percentage of the total, the trend of its change across the monocropping years was closely correlated with the total phenolic acid.

### 3.3. The Change in Microbial Community in the Healthy and Diseased American Ginseng during Various Consecutive Years of Monocropping

The composition of the bacterial and fungal microbiome between healthy and diseased American ginseng differed to a greater extent with the increase in monocropping year. Previous studies also observed the different structures of rhizosphere microbiome between the diseased and healthy plants in monocultured cotton [37], hybrid bamboo (*Bambusa pervariabilis* × *Dendrocalamopsis daii*) [38], and monkshood (*Aconitum carmichaelii Debx*.) in a 2-year monoculture field [39]. Unlike the above studies focusing on a single growing season, our finding confirmed the microbiome composition difference between healthy and diseased plants was related to the different monocropping years or monocropping history.

The difference in alpha diversity between healthy and diseased American ginseng gradually increased with the increase in monocropping years. With the increase in monocropping years, the pathogen’s abundance in the rhizosphere soil of diseased plants was higher in comparison to healthy plants. The pathogens could compete with other microorganisms for carbon sources and niches in rhizosphere soil, resulting in a decrease in microbiome diversity [40]. In addition, compared with the diseased American ginseng, more diverse root exudates secreted by the healthy plants provided a large range of carbon sources for microorganisms to enhance their diversity [41].

We found that the abundance of *Fusarium* spp. in the rhizosphere soil of diseased American ginseng was always higher than healthy plants in all the 1–4 years of monocropping history. However, the result of qPCR revealed that the abundance of *F. solani* in the rhizosphere soil of diseased American ginseng was significantly higher than the healthy plant after only 4 years of monocropping, and there was no difference in *F. oxysporum* between diseased and healthy plant. These results indicate that the *Fusarium* spp. genus included not only pathogens species but also nonpathogenic species. For example, a previous study showed that *F. oxysporum* strains isolated from the healthy plants of tomato (*Solanum lycopersicum* L.) were beneficial microbes for plant growth and resistance to pathogens [42]. In addition, we found that compared with the diseased American ginseng, the relative abundance of *Trichoderma* spp. in the rhizosphere soil of the healthy plant increased consistently across different monocropping histories. The *Thichoderma* spp. includes some species as biocontrol strains, which compete with pathogens in soil [43]; it uses enzymes to dissolve the pathogen’s cell wall, including cellulase, β-1,3-glucanase, and chitinase [44]. Meanwhile, the defense enzymes in plants, such as peroxidase (POD), phenylalanine ammonia lyase (PAL), and polyphenol oxidase (PPO), can be induced by *Trichoderma* spp. to resist the pathogen invasion and reduce the disease occurrence [45]. *Trichoderma* spp. also released ammonia, siderophores, auxin, and some secondary metabolites, which could promote the growth of plants [46,47].

### 3.4. The Phenolic Acid Suppressed the Growth of Pathogen

The increased occurrence of *F. solani* was associated with the decrease in total phenolic acid and p-coumaric acid. A similar finding was also reported previously that compared to the soil without a history of American ginseng cropping, the content of phenolic acid in the rhizosphere soil after 3 years of monocropping decreased, while pathogens, such as *Helicoma* spp., *Cyberlinkera* spp., and *Saitozyma* spp., increased [24]. Meanwhile, p-coumaric acid could also inhibit the germination of spores and sporulation of *F. oxysporum* [48]. Through bioassay, we found that the total phenolic acid at 80–100 mg·L^−1^ and p-coumaric acid at 40–100 mg·L^−1^ suppressed the growth rate of *F. solani* in vitro. The *Fusarium* wilt of cucumber (*Cucumis sativus* L.) was reduced by applying phenolic acid to soils, and with the increased concentration, the inhibition effect was enhanced [49]. Another study found that the mycelial growth and sporulation of *F. oxysporum*, the pathogen isolated from the rhizosphere soil of watermelon (*Citrullus lanatus* L.), was inhibited by cinnamic acid at concentrations between 0 and 1600 mg·L^−1^ [50].

## 4. Materials and Methods

### 4.1. Sample Collection

The long-term field experiment was conducted at Weihai city of Shandong Province, China (37°2′ N, 122°1′ E), using a split-plot design with American ginseng monocropping history (1–4 monocropping years) as the main plot and root disease infection as the subplot under 3 blocks (replicates). Samples were collected in different monocropping years. Plant roots were carefully dug out and shaken off to remove the soil that was not closely bound to the root. The soil that was tightly bound to the root was considered the rhizosphere soil. In each plot, 24 plants were sampled, and their rhizosphere soil was pooled as one sample. Finally, the 8 treatments included 1-year monocropping healthy rhizosphere soil (1Y-H), 1-year monocropping diseased rhizosphere soil (1Y-D), 2-year monocropping healthy rhizosphere soil (2Y-H), 2-year monocropping diseased rhizosphere soil (2Y-D), 3-year monocropping healthy rhizosphere soil (3Y-H), 3-year monocropping diseased rhizosphere soil (3Y-D), 4-year monocropping healthy rhizosphere soil (4Y-H), and 4-year monocropping diseased rhizosphere soil (4Y-D). Rhizosphere soils used for the determination of phenolic acid content were stored at 4 °C, and the 0.25 g of soil was stored at −80 °C to extract soil DNA.

### 4.2. The Extraction and Measurement of Phenolic Acid

According to the method of Meng et al. [51], the phenolic acids in the rhizosphere soil were extracted. Twenty grams of air-dried rhizosphere soil was placed in 500 mL conical flasks, and 150 mL of 2 mol·L^−1^ NaOH solution was added to each flask. The flasks were left for 24 h, then shaken at 210 r·min^−1^ for 30 min, centrifuged at 8000× *g* for 10 min, and the suspension was acidified to pH 2.5 with 6 mol·L^−1^ HCl, and the equal volume of ethyl acetate was added for extraction 3 times. The supernatant was mixed and concentrated by rotary steaming. The phenolic acids were redissolved in 1–2 mL 80% methanol, then filtered (0.22 μm). The filtrate was put into the injection bottle and stored at 4 °C for measurement.

Twelve compounds of phenolic acids, including protocatechuic acid, epicatechin, p-hydroxybenzoic acid, vanillic acid, p-coumaric acid, sinapic acid, vanillin, ferulic acid, benzoic acid, salicylic acid, syringic acid, and cinnamon acid, were analyzed by HPLC. The standards of the 12 phenolic acids (Sangon Biotech, Shanghai, China) were prepared at 200 mg·L^−1^.

The Agilent 1260 infinity II HPLC (Agilent Technologies Inc., Santa Clara, California, USA) was used to detect phenolic acid extracts. A Symmetry C-18 reversed column (4.6 mm × 250 mm, 5 µm) was installed in the HPLC, and the 5 µL sample was added into the column at 35 °C. The chromatograms of the sinapic acid, benzoic acid, salicylic acid, and ferulic acid were measured at 230 nm, and the chromatograms of the other phenolic acids were measured at 280 nm [52]. The 0.5% acetic water (A) and methanol (C) were used as the mobile phase, and the flow rate was set to 0.7 mL∙min^−1^ (A:C = 60:40, *V*:*V*). The separation of individual phenolic acids and total phenolic acids was performed by isometric elution and gradient elution, respectively. The following were the chromatographic conditions: First, 0–10 min, 95–80% A and 5–20% C; second, 10–13 min, 80–65% A and 20–35% C; then 13–33 min, 65–50% A and 35–50% C; and then 33–43 min, 50–5% A and 50–95% C; final, 43–53 min, 5–95% A and 95–5% C [53].

### 4.3. Soil DNA Extraction and Sequencing

The DNA was extracted by using DNeasy PowerSoil DNA Elution kit (Qiagen, Germantown, Maryland, USA). The extraction was carried out from 0.25 g rhizosphere soil samples based on the manufacturer’s instructions. The DNA integrity was detected by 1% agarose gel electrophoresis. The universal primers 341F/806R (341F: 5′-CCTAYGGGRBGCASCAG-3′/806R: 5′-GGACTACNNGGGTATCTAAT-3′) were used to amplify the V3-V4 region of the bacterial 16S rRNA gene, and the universal primers ITS1F/ITS2R (ITS1F: 5′-CTTGGTCATTTAGAGGAAGTAA-3′/ITS2R: 5′-GCTGCGTTCTTCATCGATGC-3′) were used to amplify the fungal ITS region. The PCR product was sequenced by the Illumina Novaseq 6000 sequencing system (Illumina, Santiago, CA, USA). The rest of the extracted DNA was stored at −20 °C for quantitative PCR (qPCR) analysis.

The primer removal, quality filtering, denoising, splicing, chimera removal, and singleton removal were performed using DADA2 from QIIME2 (https://view.qiime2.org) (accessed on 22 May 2023) [54]. The amplicon sequence variants (ASVs) at the 99% similarity level were obtained by clustering the clean sequences [55]. Then the ASV sequences were aligned using the QIIME2 feature-classifier plugin to a pre-trained UNITE (release 8.0) [56] and Greengenes (release 13.8) [57] databases. Finally, at each level of classification—domain, phylum, class, order, family, genus, and species—the community composition of each sample was established. The QIIME2 feature-table plugin was used to filter any contaminating mitochondrial and chloroplast sequences.

### 4.4. The qPCR Analysis

The abundance of *Fusarium solani*, *Fusarium oxysporum*, *Alternaria alternata,* and *Ilyonectria destructans* was quantified by qPCR in a C1000 thermal cycler with a CFX96 real-time system (Bio-Rad Laboratories, Hercules, California, USA) to quantify the abundance of *Fusarium solani*, *Fusarium oxysporum*, *Alternaria alternata,* and *Ilyonectria destructans*. A total of 1 µL of DNA template, 10 µL of 2×SYBR Premix Ex Taq (Accurate Biology, Changsha, China), 8 µL of sterile distilled water, and 0.5 µmol·L^−1^ of each primer, including *F. solani* (ITS1F-F: 5′-CTTGGTCATTTAGAGGAGTAA-3′, AFP346-R: 5′-GGTATGTTCACAGGGTTGATG-3′) [58], *F. oxysporum* (ITS1F-F: 5′-CTTGGTCATTTAGAGGAAGTAA-3′, AFP308-R: 5′-CGAATTAACGCGAGTCCCAAC-3′) [46], *A. alternata* (Alt-F: 5′-TGTCTTTTGCGTACTTCTTGTTTCCT-3′, Alt-R: 5′-CGACTTGTGCTGCGCTC-3′) [59], and *I. destructans* (Ily-F: 5′-GCTACCCTATAGCGCAGGTG-3′, Ily-R: 5′-CCGTACTGGCTGAAGAGTCA-3′) [59], were added to a total volume of 20 µl for the qPCR. The qPCR thermal conditions were 3 s at 95 °C, followed by 40 cycles of 95 °C for 5 s, 60 °C for 30 s, followed by melting curve analysis after the last cycle. The target fragment was cloned on a plasmid to prepare standards for qPCR. The standard curve was generated via 10 times gradient dilution of the plasmid DNA inserted with specific fragments according to the above protocol, and the efficiencies of the qPCR were 81.90–106.50%, and for all the standard curves, the coefficient of determination (R2) was >0.90.

### 4.5. Effects of Exogenous Phenolic Acid Concentration on Pathogen Suppression

Based on Durairaj’s pathogen list and methodology [60], American ginseng with root rot was collected from the fields in order to isolate pathogenic fungi. Pathogenicity tests were performed by inoculating the isolated strain on the puncture site of the surface of healthy American ginseng root. It was observed that the healthy American ginseng root exhibited symptoms similar to those on root rot plants observed in the field after inoculation. Finally, the isolates were identified as strains of *Fusarium solani* by morphological and molecular identification and were considered the target pathogens in this study.

According to the type and content of phenolic acids in the rhizosphere soil determined by HPLC in this study, total phenolic acids and p-coumaric acid were used to test their concentration on pathogen suppression. The solutions were added to PDA medium after being sterilized with a 0.22 μm filter membrane (Sangon, Shanghai, China). The total phenolic acid (1–2 years and 3–4 years) were at final concentrations of 15, 35, 60, 80, and 100 mg·L^−1^ (the 35 mg include 1.5 mg protocatechin, 3 mg p-hydroxybenzoic acid, 0.5 mg epicatechin, 2.5 mg vanillic acid, 3 mg syringic acid, 1.5 mg vanillin, 15 mg p-coumaric acid, 1.5 mg sinapic acid, 3 mg ferulic acid, 1 mg benzoic acid, 2.5 mg salicylic acid, and 0.2 mg cinnamic acid, and the other concentrations increase or decrease proportionally). The p-coumaric acid was at final concentrations of 5, 10, 20, 40, 80, and 100 mg·L^−1^. Mycelial discs (5 mm diameter each) of pure cultures of *F. solani* were incubated on PDA plates with various concentrations of phenolic acid at 25 °C for 6 days. Colony growth was evaluated by measuring the diameters based on the “cross” method [61].

### 4.6. Statistical Analysis

The core-diversity plugin within QIIME2 was used to calculate the Shannon index. Beta-diversity analysis of fungal and bacterial microbiomes in the rhizosphere soil was performed using the Bray–Curtis distance according to the “Vegan” package in R (version 4.2.1) [62]. Variable important in projection (VIP) from “mixOmics” in R (version 4.2.1) [63] and random forest (version 4.2.1) [64] was used to measure the importance of individual phenolic acids, determining the variation between experimental treatments. The relationship between the content of each phenolic acid and the abundance of pathogenic microorganisms was assessed by the Pearson correlation. SPSS 25.0 software (SPSS Inc., Chicago, IL, USA) was used to perform a single-factor analysis of variance (ANOVA) and Duncan multiple interval test (*p* < 0.05), and the homogeneity of variance was checked before statistical analysis.

## 5. Conclusions

Our results demonstrate that the pathogenic infection of American ginseng significantly influenced the secretion of phenolic acids and the assembly of the rhizosphere microbiome and that the monocropping history played a central role in modulating the interactions between disease infection, root exudation, and the diversity and structure of the microbial community in the rhizosphere soil.

## Figures and Tables

**Figure 1 plants-12-02993-f001:**
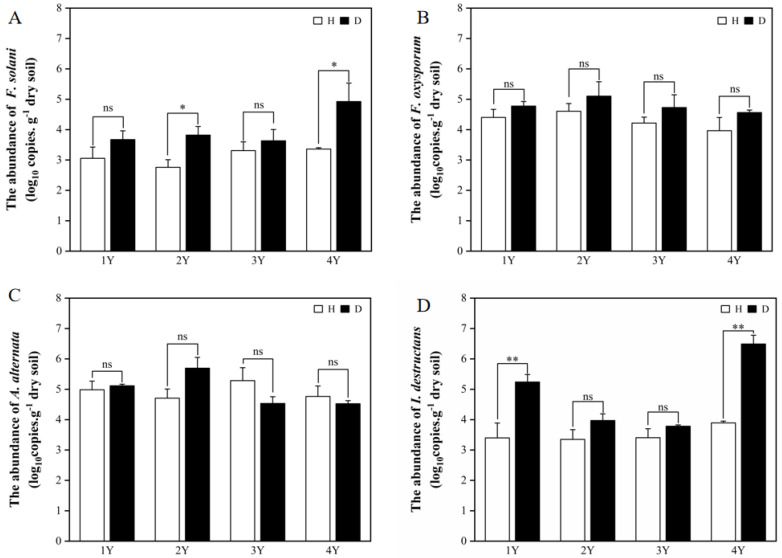
The abundance of *F. solani* (**A**), *F. oxysporum* (**B**), *A. alternata* (**C**), and *I. destructans* (**D**) in the rhizosphere soils of the healthy (H) and diseased (D) American ginseng in 1–4 years of continuous monocropping (1Y, 2Y, 3Y, and 4Y). Error bars indicate the standard errors. ns indicates no significant difference at *p* < 0.05 according to Duncan’s multiple range test, * and ** indicate significant differences at *p* < 0.05 and *p* < 0.01, respectively.

**Figure 2 plants-12-02993-f002:**
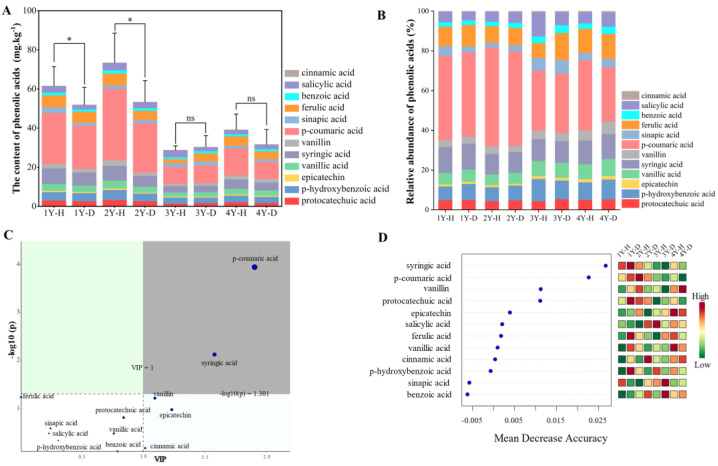
The phenolic acids content (**A**) and composition (**B**), the VIP of PLS-DA (**C**), and the random forest analysis (**D**) of phenolic acids in the rhizosphere soil of the healthy (H) and diseased (D) American ginseng in 1–4 years of monocropping (1Y, 2Y, 3Y, and 4Y). Error bars indicate the standard errors. ns indicates no significant difference at *p* < 0.05 according to Duncan’s multiple range test, * indicate significant differences at *p* < 0.05.

**Figure 3 plants-12-02993-f003:**
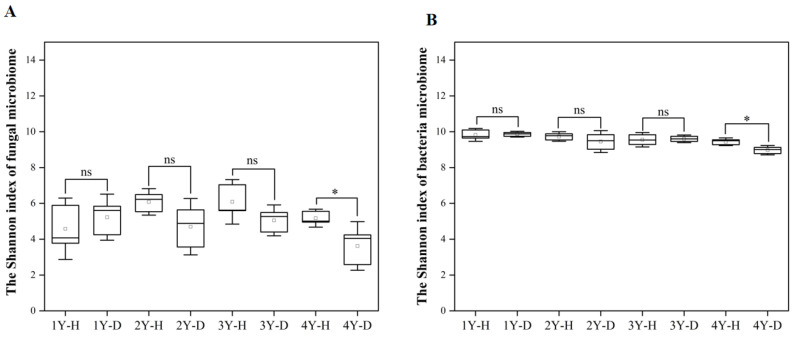
The Shannon index of fungal (**A**) and bacterial (**B**) microbiomes in the rhizosphere soil of the healthy (H) and diseased (D) American ginseng in 1–4 years of continuous monocropping (1Y, 2Y, 3Y, and 4Y). Error bars indicate the standard errors. ns indicates no significant difference at *p* < 0.05 according to Duncan’s multiple range test, * indicate significant difference sat *p* < 0.05.

**Figure 4 plants-12-02993-f004:**
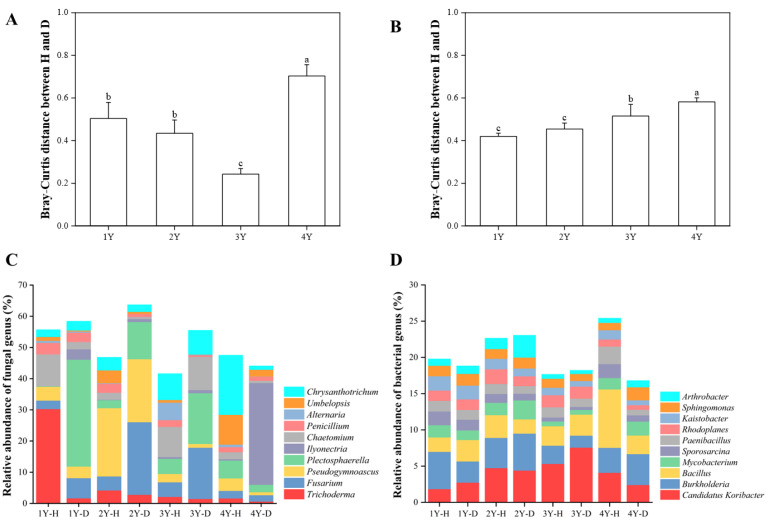
The Bray–Curtis distance of fungal (**A**) and bacterial (**B**) microbiomes between healthy and diseased American ginseng when grown in soils with different monocropping years; and the relative abundance of the main fungal (**C**) and bacterial (**D**) groups present in the rhizosphere soil of the healthy and diseased American ginseng in 1–4 years of continuous monocropping (1Y, 2Y, 3Y, and 4Y) at the genus level; only the top ten genera in relative abundance are displayed. Different letters above the bars represent significant differences at *p* < 0.05 according to Duncan’s multiple range test.

**Figure 5 plants-12-02993-f005:**
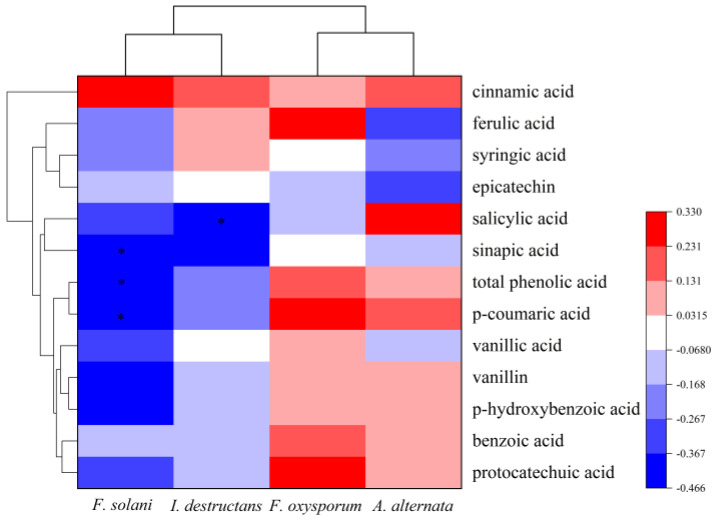
The correlations between phenolic acids with pathogens abundance. * indicates a significant difference at *p* < 0.05.

**Figure 6 plants-12-02993-f006:**
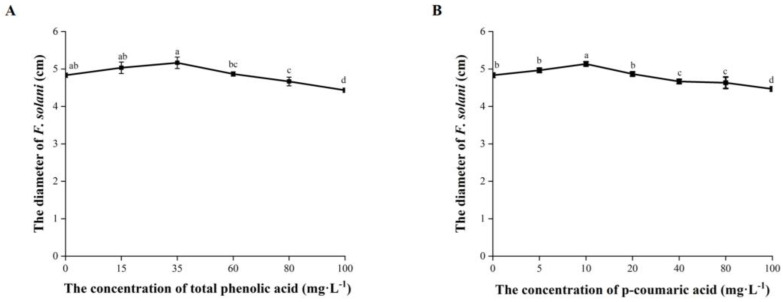
Effects of total phenolic acid and p-coumaric acid on the mycelial growth of *F. solani* (**A**,**B**) at different concentrations. Error bars indicate the standard errors. Different letters above the bars within the same phenolic acid represent significant differences at *p* < 0.05 according to Duncan’s multiple range test.

## Data Availability

All sequencing raw data were uploaded to the Genome Sequence Ar-chive (GSA) database.

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
