# Peer review of "Rhizosphere Microbiome and Phenolic Acid Exudation of the Healthy and Diseased American Ginseng Were Modulated by the Cropping History"

_plants, 2023, doi:10.3390/plants12162993_

Round 1

Reviewer 1 Report

Dear Authors you have good work - Rhizosphere Microbiome and Phenolic Acid Exudation of the Healthy and Diseased American Ginseng were Modulated by the Cropping History.

The statement you wrote in abstract, some where differ from the result. Things should be crystal clear from the abstract the effect of phenolic acid on rhizospheric microflora. It will be better to mention a particular phenolic acid responsible for rhizospheric microbiome dynamics.

There are some sentences which are not clear eg. Lines 69-72.

Conclusion; pathogenic infection on American ginseng could potentially modulate the secretion of phenolic acids, it may be possible that rhizospheric microbiome (including beneficial + pathogens) modulate the secretion of diverse phenolic acids?? 

You may also consider the following refernces;

Bao L, Liu Y, Ding Y, Shang J, Wei Y, Tan Y, Zi F. Interactions Between Phenolic Acids and Microorganisms in Rhizospheric Soil From Continuous Cropping of Panax notoginseng. Front Microbiol. 2022 Feb 24;13:791603. doi: 10.3389/fmicb.2022.791603.

Goodwin PH. The Rhizosphere Microbiome of Ginseng. Microorganisms. 2022 Jun 2;10(6):1152. doi: 10.3390/microorganisms10061152. 

Authors should check the language throughout the manuscript for minor English grammar /sentence correction.

For eg.

Lines 11-14, the sentence is too long, break it to make simple and informative.

Line 23,  were more abundant for the, it should be....... were more abundant in the.

Line 35, which quality?

Author Response

Review1:

Q1. The statement you wrote in abstract, some where differ from the result. Things should be crystal clear from the abstract the effect of phenolic acid on rhizospheric microflora. It will be better to mention a particular phenolic acid responsible for rhizospheric microbiome dynamics.

Reply: we have rewritten the whole abstract, and highlighted the effect of p-coumaric acid on pathogenic microbes and microbiome structure. Please see the Abstract in revised manuscript.

Q2. There are some sentences which are not clear eg. Lines 69-72.

Reply: we have rewritten the sentence as

“For American ginseng, the growth of the pathogen species Rhizoctonia solani was promoted by the low-level exudation of phenolic acids, e.g. p-coumaric acid, syringic acid, and vanillic acid, but suppressed by the higher concentrations of phenolic acid [23].” Please see line 81-86 in the revised manuscript with track changes.

Q3. Conclusion; pathogenic infection on American ginseng could potentially modulate the secretion of phenolic acids, it may be possible that rhizospheric microbiome (including beneficial + pathogens) modulate the secretion of diverse phenolic acids??

Reply: we agree with the reviewer’s comments, and have rewritten the conclusion to highlight the interactions between phenolic acids and microbiome in the rhizosphere. Please see the Conclusion in revised manuscript.

Q4. You may also consider the following refernces;

Bao L, Liu Y, Ding Y, Shang J, Wei Y, Tan Y, Zi F. Interactions Between Phenolic Acids and Microorganisms in Rhizospheric Soil From Continuous Cropping of Panax notoginseng. Front Microbiol. 2022 Feb 24;13:791603. doi: 10.3389/fmicb.2022.791603.

Goodwin PH. The Rhizosphere Microbiome of Ginseng. Microorganisms. 2022 Jun 2;10(6):1152. doi: 10.3390/microorganisms10061152.

Reply: we agree with the reviewer, and have added the advised two references into the revised manuscript as reference [6] and [7].

Q5. Comments on the Quality of English Language. Authors should check the language throughout the manuscript for minor English grammar /sentence correction. For eg. Lines 11-14, the sentence is too long, break it to make simple and informative.

Reply: the sentence has been rewritten as:

“The infection of soil-borne diseases has the potential to modify root exudation and the rhizosphere microbiome. However, the extent to which these modifications occur in various monocropping histories remains inadequately explored. This study sampled the healthy and diseased American ginseng (Panax quinquefolius L.) plants under 1-4 years monocropping, and analysed….” Please see line 11-15 in the revised manuscript with track changes.

Q6. Line 23,  were more abundant for the, it should be....... were more abundant in the.

Reply: “…were more abundant for…” has been replaced by “…were more abundant in…”

Q7. Line 35, which quality?

Reply: the first sentence in the Introduction has been rewritten as

“American ginseng (Panax quinquefolius L.) is one of the most important medical plants in the world, and has been widely cultivated under the monocropping to increase the productivity. However, this practice has also showed negative impacts on the American ginseng's medicinal quality [1]”

Reviewer 2 Report

The comparison between heathy and diseased American ginseng over a 4-year field growth period is a good study to understand the real-world microbiome interactions. This kind of a study is useful in raising healthy plants for medicinal value. The manuscript needs some corrections, such as those mentioned below.

Lines 11-14: The first line of the Abstract needs to be improved. It does not flow with the rest of the Abstract.

Introduction:

Line 33-34: monocropping of Ginseng is on the rise for what increased demands? Give some details of the use and hence the reason for increased demand.

Line 72: with high concentrations suppressing them.

Line 76: and the pathogens’ occurrence

In the results section, in some places the amount of the phenolics studied are mentioned as higher or lower. It will be better to mention a % difference as higher or lower.

Line 230: diseased plants was larger

Line 240: Panax – check all the scientific names mentioned for the full form in the first mention and then the shortened version.

Lines 240-242: p-coumaric acid is an important phenolic acid

Line 305: Correct the samples collection date. If 1-4 monocropping years were considered, then there must be 4-yearly sample collection, right?

Lines 385: It was observed that the …

Line 410: Pearson correlation

The manuscript seems to be written well, except for some minor errors that can be easily corrected.

Author Response

Reviewer2

Q1. Lines 11-14: The first line of the Abstract needs to be improved. It does not flow with the rest of the Abstract.

Reply: we have rewritten the whole abstract, the first sentence in the Abstract is as:

“The infection of soil-borne diseases has the potential to modify root exudation and the rhizosphere microbiome. However, the extent to which these modifications occur in various monocropping histories remains inadequately explored.”

Q2. Introduction: Line 33-34: monocropping of Ginseng is on the rise for what increased demands? Give some details of the use and hence the reason for increased demand.

Reply: we have rewritten the first sentence in Introduction as

“American ginseng (Panax quinquefolius L.) is one of the most important medical plants in the world, and has been widely cultivated under the monocropping to increase the productivity. However, this practice has also showed negative impacts on the American ginseng's medicinal quality [1].”

Q3. Line 72: with high concentrations suppressing them.

Reply: we have rewritten the sentence as

“For American ginseng, the growth of the pathogen species Rhizoctonia solani was promoted by the low-level exudation of phenolic acids, e.g. p-coumaric acid, syringic acid, and vanillic acid, but suppressed by the higher concentrations of phenolic acid [23].” Please see line 81-86 in the revised manuscript with track changes.

Q4. Line 76: and the pathogens’ occurrence

Reply: “pathogens occurrence” has been replaced by “pathogens’ occurrence”

Q5. In the results section, in some places the amount of the phenolics studied are mentioned as higher or lower. It will be better to mention a % difference as higher or lower.

Reply: we have added the % values when describing the amount of the phenolics, please see line 128-136 in the revised manuscript.

Q6. Line 230: diseased plants was larger

Reply: we have rewritten the sentence as

“The phenolic acid content in the healthy plants was larger than diseased plants after short-term monocropping ….”

Q7. Line 240: Panax – check all the scientific names mentioned for the full form in the first mention and then the shortened version.

Reply: “P.” has been used in the whole manuscript to replace “Panax” after mentioned the full name firstly.

Q8. Lines 240-242: p-coumaric acid is an important phenolic acid

Reply: “was” has been replaced by “is”.

Q9. Line 305: Correct the samples collection date. If 1-4 monocropping years were considered, then there must be 4-yearly sample collection, right?

Reply: we have rewritten the sentence as

“Samples were collected in different monocropping years.”

Q10. Lines 385: It was observed that the …

Reply: “Then we found that…” has been replace by “It was observed that…”

Q11. Line 410: Pearson correlation

Reply: “Person correlation” has been replace by “Pearson correlation”